# Using Social Marketing to Promote COVID-19 Vaccination Uptake: A Case Study from the “AUBe Vaccinated” Campaign

**DOI:** 10.3390/vaccines11020459

**Published:** 2023-02-16

**Authors:** Marco Bardus, Sara A. Assaf, Carine J. Sakr

**Affiliations:** 1Institute of Applied Health Research, College of Medical and Dental Sciences, University of Birmingham, Birmingham B15 2TT, UK; 2Department of Health Promotion and Community Health, Faculty of Health Sciences, American University of Beirut, Beirut 1107 2020, Lebanon; 3Employee Health Unit, Faculty of Medicine American University of Beirut, Beirut 1107 2020, Lebanon

**Keywords:** COVID-19, vaccination, social marketing, demand creation, health communication, branding, university, Lebanon

## Abstract

The availability of and access to COVID-19 vaccines has been challenging in many low- and middle-income countries (LMICs), coupled with mistrust in public health organizations instigated by misinformation and disinformation diffused by traditional and social media. In the Spring of 2021, the American University of Beirut (AUB) in Lebanon spearheaded a nationwide vaccination drive with the ambitious goal of vaccinating its entire community by the beginning of the academic year 2021–2022, as the campus was due to be opened only to vaccinated individuals. This case study outlines the development, implementation, and evaluation of a social marketing campaign to encourage COVID-19 vaccinations among members of the AUB community, comprising students, faculty, staff, and dependents. Following French and Evans’ 2020 guidelines, we implemented an evidence-based and co-designed strategy to maximize the availability and facilitate vaccine access. The campaign used a mix of methods to convince the segments of the population to receive their shots before accessing campus, resulting in a 98% uptake among the community segments within three months (July–September 2021). In this case study, we reflect on the experience and share suggestions for future research and applications that other higher education institutions could use to address similar problems.

## 1. Introduction

The coronavirus disease 2019 (COVID-19) pandemic had a devastating impact on economies, education, health care, and social aspects, and it posed a severe threat to human health globally. However, these implications are more profound in low- and middle-income countries (LMICs), where healthcare capacity is often constrained, and citizens are already battling food, security, economic, and health crises [1,2,3]. The mortality toll from COVID-19 has been four times greater in lower-income nations than in higher-income countries, according to a new Oxfam research published two years after the World Health Organization declared COVID-19 a pandemic [4]. Despite substantial research, no treatment for SARS-CoV-2 infection has yet been consistently successful in controlled studies. Global immunization against SARS-CoV-2 is thus the only possibility for a breakthrough in the battle against this emerging virus [5]. Even after developing safe and effective vaccinations, LMICs continued to face challenges to care due to inequitable access and vaccine hesitancy [6]. Following the release of the first COVID-19 vaccine batch in March 2021, many LMICs faced a demand higher than the available supply, as 85% of global vaccine doses were delivered to high- and upper-middle-income countries [7]. The inequalities were even starker in some areas of the African continent, where most countries had administered doses to less than 1% of their population [7]. In late 2021, only about 25% of people living in LMICs have received at least one dose [8]. Furthermore, as time passed, vaccine hesitancy became an essential factor in achieving global population protection [6]. A recent meta-analysis and systematic review showed that LMICs had a pooled COVID-19 vaccine acceptance rate of 58.5% (33 studies) and a pooled vaccine hesitancy rate of 38.2% (32 studies) [6].

Vaccine hesitancy has been a global public health challenge since before COVID-19, negatively affecting child and adult immunizations. A review of systematic reviews on vaccine hesitancy identified some common strategies to address this problem [9]. Strategies may include a combination of individual-level, face-to-face [10] or community-wide mobilization efforts [11] to dispel myths and encourage vaccination demand using social and traditional mass media channels. For example, a recent US-based campaign promoted COVID-19 vaccine uptake among an underrepresented segment of the population by disseminating short medical videos (<1 min) as advertisements on Facebook and using geo-targeting to reach areas with high COVID-19 death rates [12]. The campaign reached about 9.9 million views demonstrating the feasibility of a targeted social media-based communications [12]. Another study from China showed a successful COVID-19 vaccination strategy, including vaccine deployment and promotion to increase availability, accessibility, and vaccine publicity [13]. Yet, belief and behavior change remain an important challenge to pro-vaccination campaigns. Moreover, the evidence base is mainly generated in the Americas and high-income countries [9,14], and what makes strategies to address vaccine hesitancy effective needs to be clarified [9].

Strategic approaches to complex public health problems include social marketing, a discipline and field of study focused on promoting ideas, changing beliefs, and ultimately influencing behaviors of individuals and communities for the greater social good. Social marketing is a practice-based framework guided by ethical principles, integrating “research, best practice, theory, audience, and partnership insight, to inform the delivery of competition-sensitive and segmented social change programs that are effective, efficient, equitable and sustainable” [15]. Social marketing has been successfully employed to address vaccine hesitancy in various contexts and settings [16]. For example, it has been used in childhood immunization programs (e.g., general immunization and HPV vaccines) that cannot be addressed by one-size-fits-all solutions preceding the parental refusal and misconceptions about vaccine consequences [17]. For example, social marketing campaigns increased immunization coverage in Colombia by 40% in 5 years (1979–1984) and the Philippines by 24% in one year (1990) [18]. More recently, social marketing campaigns in rural areas in the US increased HPV uptake by 34% in 3 months [19,20]. A more recent study reported the successful implementation of a social marketing campaign to promote seasonal flu vaccination at a university in Hong Kong [21]. Another case study described a comprehensive social marketing plan that could be utilized to address COVID-19 vaccine hesitancy among senior citizens of rural India [22].

Building on this new evidence base, social marketing has been recommended to create demand for COVID-19 vaccines [23] while addressing inequitable access to vaccines up front, as outlined in a seminal paper by French and colleagues [24]. The strategic importance of co-producing solutions with the target populations is critical to delivering successful interventions. How does social marketing work? According to French and Evans [23], exposure to a COVID-19 vaccine campaign produces changes in awareness, knowledge, attitude, and beliefs, which directly affect the intention to vaccinate, which in turn has a direct effect on vaccination uptake. At the same time, social marketing-based campaigns aimed at improving vaccine uptake should account for the perceived social, structural, and economic influences of the behavior [23]. Yet, more is needed to know about how strategies are implemented, as documentation of processes and methodologies on this aspect needs to be improved.

To fill the research and implementation gap on the strategic use of social marketing to promote COVID-19 vaccine uptake, we present our case study, which describes the process and outlines the results of a campaign targeting an academic community. This case study aims to (a) demonstrate the feasibility and utility of social marketing in designing a campaign to promote COVID-19 vaccination uptake and (b) evaluate its impact to provide insights that policymakers and social marketers could use to develop similar interventions.

## 2. Materials and Methods

We adopted a practice-oriented research perspective described in the “Case Study Methodology in Business Research” book [25]. Practice-oriented research aims to describe an intervention and reflect on its findings. These findings, in turn, contribute to enhancing practitioners’ knowledge and fostering the development of the field. As such, we do not intend to contribute to theory development or any other generalizations [25]. The structure of a practice-oriented case study follows the one of a descriptive, observational study. In this section, we discuss the context and methods and elaborate on the results in the Section 4.

### 2.1. The Context

In Lebanon, authorities announced the country’s first confirmed case of COVID-19 on 21 February 2020 [26], and the first confirmed COVID-19-related death less than two months later, on 10 March [27]. Between February and December 2020, 177,996 confirmed cases of COVID-19 and 2878 deaths; in the first 20 days of 2021, 77,704 new COVID-19 cases were recorded, representing an average of 3885 new cases per day [28]. As a result, the healthcare system was crippled, as Intensive Care Units (ICU) were at 91% occupancy across the country and 100% in Beirut [28].

In February 2021, the local Ministry of Public Health (MOPH) launched a national deployment and vaccination plan supported by various United Nations agencies, the World Bank, and the World Health Organization. The MOPH received the first batch of COVID-19 vaccines on 13 February 2021, with aid from the World Bank [29], which mandated the government to promote an equitable access to vaccines. To address the limited COVID-19 vaccines supply, a risk-and age-based approach was adopted to prioritize target groups and ensure an efficient and timely vaccine distribution [30]. High-priority groups included frontline health workers, the elderly, and those with co-morbidities [30]. However, the deployment of vaccines lagged, owing to inefficiencies and a lack of coordination efforts. As of March 2021, only about 6% of the population residing in Lebanon had been vaccinated, with only 33% of the population vaccine-favorable [31]. Nevertheless, some non-representative surveys showed more extensive acceptance among university students [32].

In April 2021, to overcome the challenges faced by the national campaign, the American University of Beirut (AUB) spearheaded other private institutions in Lebanon and committed to privately purchasing and administering 90,000 doses of Pfizer BioNTech to cover 45,000 individuals, including students, faculty, staff, and their dependents, in addition to vulnerable populations who had no access to vaccines [33]. Hence, an AUB-led COVID-19 vaccination campaign was initiated to address the local vaccination demand in the presence of a low vaccination supply nationally and the urgent need to vaccinate the AUB community to safely open its campus by the start of the new academic year 2021–22. According to the official statistics, in March 2021, the AUB community included 15,369 individuals, 1324 faculty members, 3285 staff, and 9495 students (7794 undergraduate and 1701 graduate) [34]. In addition, the extended community encompassed dependents of AUB faculty, staff, students, alums, their close and extended family, and vulnerable populations reached through academic partnerships, volunteering, and humanitarian action.

An institution-wide working group was formed to manage the vaccination drive (the “Vax-WG” committee), including representatives from the AUB Medical Center, Pharmacy, Registrar, Nursing, Quality and Risk Management, Office of Student Affairs, Human Resources, Communication Department, Security Office, Information Technology, and members of the Faculties of Nursing, Medicine, and Health Sciences. The third author co-chaired the Vax-WG committee, while the first author, an expert in health and risk communication and social marketing, co-developed the overarching social marketing strategy with the support of other colleagues from the University.

In this paper, we present the strategy and reflect on its implementation over the summer of 2021 (July 2021 to September 2021). At the beginning of 2022, AUB had access to new vaccines and started offering third doses of Pfizer and Moderna (Spikevax), which were not part of the initial communication strategy.

### 2.2. Planning Framework

In developing the vaccination campaign, we followed a 10-step social marketing process outlined by French and colleagues in their guidelines for developing a COVID-19 vaccination uptake [24]. The process we followed is depicted in Figure 1 below. Our strategy was based on evidence, gaining insight from the target population, and receiving constant feedback to adjust the tactics utilized.

#### 2.2.1. Behavior Change Planning

Our campaign followed clear behavioral goals set by the institution, was evidence-based, and rooted in behavioral theories. According to AUB leadership, the purpose of the campaign was to vaccinate at least 95% of the AUB community to build “herd immunity” and allow a safe return to on-campus learning by the start of the new academic year 2021–2022 (i.e., the last week of August 2021). Following a national containment strategy, access to campus would have been restricted to vaccinated individuals. Based on our situation analysis, we defined two main strategic goals: (1) facilitate access to vaccines and (2) create vaccine demand. Specific behavioral objectives were to (1) register for the vaccine using the Ministry of Public Health COVAX platform—the only way for anyone residing in Lebanon to receive a vaccination; and (2) receive at least one dose of the vaccine to access campus.

Behavior change planning was informed by an anonymous web-based survey we conducted among the AUB community at the beginning of June 2021. The survey was aimed to assess the vaccination status, the intention to receive the vaccine, the reasons for delaying vaccination, and the preferred communication channels. We evaluated the intention to receive a vaccine through a 7-point likelihood scale (7 = definitely yes; 1 = definitely not) as a response to the question “How likely would you get the Pfizer BioNTech if this was offered to you next week?” as carried out in recent studies [35,36]. In line with the literature on the vaccine hesitancy [16,32], we defined three groups: those who responded “definitely,” “very likely,” or “somewhat likely” were considered “in favor of vaccine”; those who answered “somewhat unlikely,” “very unlikely,” or “unsure” were considered “hesitant”; those who responded “definitely not” were considered “vaccine-resistant.”

#### 2.2.2. Audience Targeting and Segmentation

The campaign focused on the core AUB community, which was segmented according to the type of membership (faculty, staff, and students), and the need to access campus as follows: (1) those who needed to access the campus during the summer semester (approximately 4000+ students and 1000+ staff); (2) those who needed to access campus in the fall semester (9400+ students and 4500+ staff). The third group, including recent graduates, long-time alumni, dependents, and some vulnerable populations, followed the first two priority groups.

Another segmentation variable was the intention to receive the vaccine. In line with strategic goal #1 (facilitate vaccine access), we prioritized those in favor of the vaccine, as they were ready and willing to adopt the desired behavior. The “vaccine-hesitant” were targeted as part of strategic goal #2 (demand creation).

We also acknowledged that some people might have had valid medical reasons for not taking vaccines. People could request a medical waiver which a central medical committee reviewed. If a waiver was granted, they were counted as compliant.

#### 2.2.3. Competition Analysis

Our analysis was based on the information collected through the formative survey and included the factors that hindered vaccine uptake.

#### 2.2.4. Mobilization and Community Engagement

After the survey, we mobilized the Immunization Center and coordinated with the Ministry of Health, which was and still is responsible for supplying the vaccines. This was because there were limitations in vaccine availability and capacity. While the Immunization Center could administer up to 2000 vaccines per day as a walk-in service, the Ministry could not provide more than 500–800 doses per day for the AUB and the external community.

#### 2.2.5. Vaccine Demand-Building and Access Strategy

Based on the segmentation mentioned above, we started the campaign by accommodating the relatively high vaccine demand from about a third of the AUB community (based on the number of survey respondents- see below). In a later stage of the campaign, we would have addressed the vaccine-hesitant and the vaccine-resistant segments of the population.

The social offering of this campaign was the Pfizer BioNTech vaccine (product) offered for free (price) to all AUB community members. The AUB Immunization center (place) is in the heart of Ras Beirut, northwest of the Lebanese capital. The promotion strategy included a coordinated communication strategy leveraging the institutional communication channels to maximize the reach of the messages among the core AUB and external communities, following the strategic goals.

**Branding:** The Office of Communications created a logo and slogan for the “AUBe Vaccinated” campaign, used in all communication materials, encompassing the campaign website, emails, booking system, social media messages, and roll-up available on campus entrances. Branding is an effective communication strategy in the public health research and practice [37,38,39] and is generally recommended to increase the intangible mental associations with the desired behaviors and campaign materials.

**Email invitations and updates:** Based on the survey results, we used emails to inform the AUB community about the strategy, explain the logistics (how, when, and where to receive the shot), and invite them to visit the Immunization Center. The Vax-WG used weekly (and sometimes more frequent) communications with the entire AUB community. The campaign was officially launched during the third week of June 2021.

**Booking system:** Piloted and launched after the first two weeks, a booking system was developed using Microsoft Bookings. We created different pages for first- and second-dose appointments. The booking system was piloted for two weeks and adopted at the end of July 2021.

**Website:** A campaign website was created at the end of July 2021 to keep an archive of all email communications and to provide more information about the logistics (how, when, and where to receive the shot), resources addressing the common concerns about safety and efficacy concerns with frequently asked questions, and pages highlighting the vaccine benefits. The website, which is still active, also included links to book the vaccination appointments and a summary of the overarching strategy [40].

**Digital media:** Even though the AUB community expressed preference towards emails, to be more inclusive of segments of the community who were not active email users, we summarized the content of the emails into posts diffused on the institutional social media accounts on Twitter, Facebook, Instagram, and invited users to diffuse messages among the informal social networks on WhatsApp groups within departments and units. Social media messages were written in English and Arabic for broader outreach.

**Personal communications:** In addition to traditional and new media, we utilized word-of-mouth and personal contacts to encourage community members to vaccinate to be inclusive and account for segments of the population less tech-savvy. Informal and formal communications happened between line managers and employees within the university units (faculty, human resources, etc.) and with the student corpus (through the office of student affairs, student clubs).

Additionally, we leveraged the security officers at the gates, who oversee access to campus. The IT department matched the vaccination status on the students and staff database with the database allowing access to campus. As per university policy, campus access was monitored and restricted to vaccinated-only individuals (see below). As mentioned before, community members could request a medical waiver to be recorded as vaccinated. If a student or staff member was not registered to vaccinate or did not receive their shot, the security officers at the campus gates were instructed to remind them to receive their shot at the earliest opportunity.

### 2.3. Monitoring and Evaluation

We used different sources and data types to monitor and evaluate the process and impact of our campaign. We listened to the community by analyzing unsolicited and unprompted feedback provided via email, social media, and informal personal communications with students and colleagues. People had the option to email questions describing their concerns. Some even spoke over the phone with physicians who addressed their queries. We monitored the clicks on the booking system pages to understand the level of demand. We tracked the number of vaccinations provided by the Immunization Center. We used all this information to refine the strategic decisions and adjust when necessary.

## 3. Results

### 3.1. Audience Insights, Competition Analysis, Segmentation

The survey we launched before the campaign, between June and July, was completed by 5664 individuals, primarily students (75%), staff (12%), and faculty members (8%). While it was based on convenience sampling, the survey allowed us to understand the hesitancy level and how we could promote vaccine uptake effectively in our community. In line with a concurrent study [31], most of the sample was in favor of the vaccine (96%), with only 2% being considered “vaccine-hesitant” and 1% “vaccine-resistant.” Our segmentation included these attitude-intention elements and the need to access campus before the start of the academic year.

Regarding competition analysis, factors hampering vaccine uptake included beliefs about vaccine safety and efficacy based on traditional and social media misinformation. Furthermore, the main reasons for refusing or delaying the vaccine were concerns about the Pfizer vaccine (side effects, efficacy, safety) (49%), logistical issues, i.e., not knowing how to schedule an appointment, time conflicts (20%), and battling a current infection (13%). The survey also suggested the preferred communication channels for the campaign, which were emails, followed by text messages/WhatsApp, and social media.

### 3.2. Mobilization and Community Engagement

Many volunteers, including students, helped throughout the campaign. They provided support with the platform and with the registration process. We also listened to the community by collecting informal and formal feedback from students, faculty, and staff who visited the vaccination center. The Vax-WG email address collected requests and complaints about the vaccination process and policy decisions. Moreover, Vax-WG members, being part of different frontline units in the vaccination center, received comments and suggestions verbally or through their email addresses. All messages were addressed within one or two days.

### 3.3. Vaccine Demand-Building and Access Strategy

In the campaign’s first two weeks, the Immunization Center could accommodate only walk-in vaccinations and administer the limited supply of 800 vaccines per day. Thousands of AUB community members visited the Immunization Center the day following Vax-WG’s initial email outlining the vaccination plan. With more than 5000 students and staff members expressing their interest in vaccinating, this resulted in long queues and many community members expressing discontent with Vax-WG members and volunteers. Listening to this feedback urged us to create a Microsoft Bookings system to absorb and organize the high demand. At the same time, we decided to send emails using a more diluted, segmented approach. The Registrar and HR Office provided lists of students and staff who needed to access campus because of summer work activities. Social media messages were not used to limit the numbers. The Immunization Center pleaded for more vaccines from the Ministry of Health, which increased the supply by the end of July.

It took about four weeks to absorb the initial vaccine demand, including about 6000+ members of the community who needed to access campus during the summer term. However, the walk-in system, fueled by controlled email invitations, allowed us to manage the high demand until the booking system was launched (at the end of July), when the Immunization Center could accommodate about 2000–3000 appointments/week.

In the week before the start of the new academic year (i.e., the last week of August 2021), the Immunization Center increased its capacity, administering more than 8300 vaccines/week (see Figure 2 below), reaching about 75% of the AUB community. Then, we focused on the demand creation strategy, resorting to biweekly social media posts that promote the booking pages in addition to more direct email invitations and reminders. The messages we diffused aimed at changing the perceived social norms [41] by demonstrating that “more than 75% of the community has already engaged in the behavior”. Messages also highlighted the benefits of the vaccine (i.e., returning to campus, protecting each other) and the costs of the alternative behavior (inability to access campus) while respecting individuals’ voluntary decisions. The email and social media posts generated 44,000 clicks on the link, promoting the first dose of the vaccine between 20 August and 17 September 2021.

Finally, the Moodle barring policy motivated the remaining vaccine-hesitant, allowing the campaign to attain its target of reaching 98% of the core AUB community by the second week of September 2021.

## 4. Discussion

This case study reflects on our experience developing and implementing a COVID-19 vaccination campaign targeting the AUB community using the 10-step social marketing process [24] which led to a substantial increase in COVID-19 vaccination among university students and other AUB community members. As a result, the campaign reached the target in less than three months (July 2021 to September 2021). In addition, it vaccinated 98% of the core AUB community (including 1324 faculty members, 3285 staff, and 9495 students) by mid-September 2021. In the following sections, we reflect on the approach used, the insights we gained, and the strategy we implemented.

### 4.1. Behavior Change Planning

We followed the social marketing planning framework described in French et al. 2020, following the guidelines for a pre-emptive vaccination campaign [23,24]. This allowed us to approach the problem from a complex, multilevel perspective. First, we engaged our leadership to develop and enforce policies, following an upstream social marketing approach. This was coupled with discussions and arrangements at the infrastructural, through negotiations with the staff and personnel of the Medical Center, reaching a midstream level of influence. Community engagement and encouraging participation allowed us to reach individual community members (downstream) so that they could feel more compelled to receive a vaccine. Multilevel, upstream-midstream-downstream approaches are more effective than focusing on single levels of influence [42]. We did not engage with the news media as indicated in French et al.’s framework because we deemed it unnecessary, with our community being cohesive and responsive to internal communication channels. In fact, our campaign strategy was based on data, and social marketing is an evidence-based discipline and field of study, which has been recommended to create demand for COVID-19 vaccines [23]. Through our approach, we tried to maximize access to reduce inequities [24]. While we could not guarantee 100% coverage of the population, we nearly reached the target, ensuring to facilitate access to vaccines for 98% of our community.

### 4.2. Formative Research Insights, Competition Analysis, Audience Segmentation

Our strategy was based on insight from our target population, competition analysis, and segmentation. The latter is a critical element in social marketing literature [43], also applied in similar campaigns elsewhere [44]. In line with a concurrent cross-sectional study [31], most of the sample favored the vaccine, with only 2% considered “vaccine-hesitant.” As such, we focused on facilitating access to vaccines for those “in favor” through adopting walk-in vaccinations, increasing the capacity of the COVID-19 vaccination center, and developing an online booking system later to organize vaccination schedules and avoid long waiting lines. A similar method was used to segment Hong Kong university students to promote seasonal flu vaccine [21] and somehow aligned with the systematic segmentation of a national sample of Australians. The former study found that 52% of the total vaccinated individuals were from the “convinced” segment, and 35% were open to being persuaded [21].

In contrast, the audience segmentation method used by Thaker and colleagues highlighted significant sociodemographic differences across segments such as age, gender, education, income, and state of residence, that is, socioeconomic status [44]. For example, vaccine enthusiasts were more likely to be older, highly educated, and male. These findings highlight the importance of setting and segmenting the vaccination target population to establish and adopt proper campaign strategies that absorb the high demand of vaccine-favorable or enthusiasts [44].

Following the absorption of the initial demand, our campaign shifted its focus towards influential vaccine-hesitant individuals using a multipronged promotion strategy. This included different communication materials and channels. Concomitantly, low vaccine uptake in the country was fueled by a small but vocal anti-vaccine movement fueled by inaccurate and sensationalistic media reporting and social media disinformation. In addition to the pandemic, Lebanon was (and still is) vexed by multifaceted social, political, and economic crises, with the scarcity of fuel, electricity, and food and the deterioration of the Lebanese currency, which made the vaccines a low-priority issue [45]. Additionally, another monetary barrier to the behavior was the spiraling cost of fuel, which made it difficult for some members of the AUB community to travel to Beirut to receive their shot.

### 4.3. Mobilization and Community Engagement

Throughout the campaign, we maintained community engagement to understand the demand and to refine our tactics based on individual feedback via email, social media, and informal personal communications. We listened and used existing social networks and leadership structures to our advantage. The effectiveness of community engagement in the success of previous vaccination campaigns has also been proven in many settings, such as Ebola and Polio vaccination campaigns in South Asia. According to the World Health Organization’s technical advisory group on behavioral insights and sciences for the health [46], community engagement can be beneficial in pre-designing vaccination methods and messaging, sharing timely information about vaccine strategies, fostering trust, and addressing false information. For example, community engagement was essential to increase COVID-19 vaccination awareness and uptake in small communities in Pakistan [47]. In addition, researchers leveraged community leaders to improve vaccine willingness by changing residents’ behavioral intentions [47].

Following the “make every contact count” [48] principle, we tried to promote behavior at every opportunity. This strategy resulted in more than 40,000 clicks on the vaccination booking system link between 20 August and 17 September 2021. A similar scenario was seen in the study by Lee et al., as different communication strategies were used to reach more students, including emails, banner ads on the e-learning portal posters, panels, and broadcasted SMS messages [21]. Banner ads on the e-learning portal and email were the most effective communication channel, sharing more than 80% of total registrations throughout the campaign period [21].

Similarly, a social media-based campaign in the United States successfully promoted COVID-19 vaccine uptake in an underrepresented group using geo-targeting on Facebook, achieving more than 9.9 million views from targeted areas [12]. We relied on digital media channels because our target population was relatively tech-savvy, educated, and affluent, as digital media use, education, and socioeconomic status correlate with digital health literacy [49]. Thus, establishing a communication strategy tailored to the needs and characteristics of the target population is a critical factor in increasing the reach of the implemented vaccination campaigns. We tried to make the campaign materials easy to understand for a wider population. At the same time, most of our community includes students and staff with a high level of education, but we could not assume that all understood our messages. Knowing the role of functional literacy and culture in the Lebanese context [50], we chose an inclusive approach in language and media channels, according to the insight gained throughout the campaign.

### 4.4. Vaccine Demand-Building and Access Strategy

The demand-building strategy we adopted entailed a mix of nudge and shove tactics [51] to convince the minoritarian yet reactive segment of the community. First, access to campus was limited to only those vaccinated or exempt for medical reasons. This can be considered a nudge tactic, as the policy entailed a “new norm” or “change to the default” choice architecture, that is access to campus. Then, towards the end of the summer, considering the urgency of getting faculty and students vaccinated before the start of the academic year, the university leadership, the office of the Provost, and the Registrar decided to suspend access to Moodle, the institutional learning management system, for those who did not provide a vaccination certificate or a valid and approved waiver form. In addition, students’ IDs were directly linked to students’ vaccination status to ensure that unvaccinated individuals were not allowed to enter the campus. This can be considered a shove tactic as it entails punishment for those who did not change their behavior [51]. While we received some complaints about this latter decision, the Vax-WG decided to make every effort to make vaccination as easy and convenient as possible; we urged the Immunization Center to accommodate those who needed to get vaccinated (laggards) and increase capacity, offering vaccinations also during weekends.

This strategy enormously facilitated reaching our target goal of vaccinating 98% of the AUB core community on time before the start of the new academic year. This scenario was also seen in other US colleges where COVID-19 vaccination was a mandatory requirement to enter campus [52]. On a larger scale, many countries employed disincentive tactics to boost vaccination uptake, such as prohibiting unvaccinated individuals from dining indoors at restaurants or accessing clubs or public events [53]. Thus, adopting incentive mechanisms is highly recommended to enhance the uptake of COVID-19 vaccines.

This campaign taught us numerous lessons. First, an effective and efficient vaccination drive needs to rely on a whole-system approach and joint inter-departmental collaboration; with the coordination of the Vax-WG committee, the campaign could be implemented, considering the external influences and the limited resources available. Second, social listening and community engagement are fundamental tools in any social marketing and risk communication campaign [54]. These tools allow for the detection of problems and identify potential solutions. Third, we learned to be flexible and to adapt the strategy to the circumstances, especially when considering structural and economic influences [23]. Fourth, we realized the importance of combining different approaches and tactics to accommodate the needs and wants of the target population, as one size does not fit all.

### 4.5. Strengths and Limitations

The results we present here might not be generalizable to other settings since our campaign occurred in a particular community, an academic institution, and its medical center. Generally speaking, our target population represents a highly educated and relatively affluent population possessing functional, digital, and health literacies, as testified by a recent study investigating the relationship between internet use and eHealth literacy [49]. However, the approach might be replicated in similar contexts in Lebanon and abroad. Additionally, despite all our efforts, we might have yet to be able to reach out to all the constituents. For example, there might have been a few student candidates who did not enroll in our university or other employees or faculty who left their positions because of the vaccine requirement.

Despite these limitations, our approach was innovative, and we are not aware of any similar campaign with such outreach in Lebanon or the region. The local expertise in social marketing that was applied was based on several years of knowledge of co-production between the university and its extended community. Recent exemplar applications of campaigns included those developed within a master-level course on “Social marketing for public health,” targeting academic units within the same university [55,56].

From a theoretical viewpoint, we treated the introduction of the Pfizer vaccine in our community with the lens of the diffusion of innovations theory [57,58], which has been recently used to explain COVID-19 vaccine uptake in a cross-sectional study with university students in China [59]. According to this theory, the respondents to the questionnaire, representing approximately a third of the AUB community, were considered “early adopters.” We then focused on these, making it easier and more convenient for them to receive the vaccine, to generate a critical mass (“early majority”) that would motivate the remaining community segments. Eventually, the late majority and the laggards (most resistant and hesitant) could have been inspired by the perceived social norm that vaccination is a norm, and the majority is vaccinated. The “social norms” approach, when there is an absolute majority of individuals performing a behavior, is commonly used in these contexts to encourage those more resistant to change [41]. This approach could partly explain the success of our campaign.

## 5. Conclusions and Recommendations

Social marketing was a valuable framework to plan, implement, and evaluate a COVID-19 vaccination drive targeting a university community in low-resource settings. This approach and methodology could be used to plan similar campaigns in other similar contexts, in Lebanon or abroad. To effectively implement and scale up a similar campaign, a multidisciplinary, inter-departmental, and intra-organizational entity (e.g., a working group or committee) should be created. In addition, clear roles and responsibilities should be shared among the members and organizations participating in a vaccination drive to maximize efficiency and effectiveness. Finally, social listening and community engagement should be carefully embedded in social marketing planning to maximize vaccine uptake.

## Figures and Tables

**Figure 1 vaccines-11-00459-f001:**
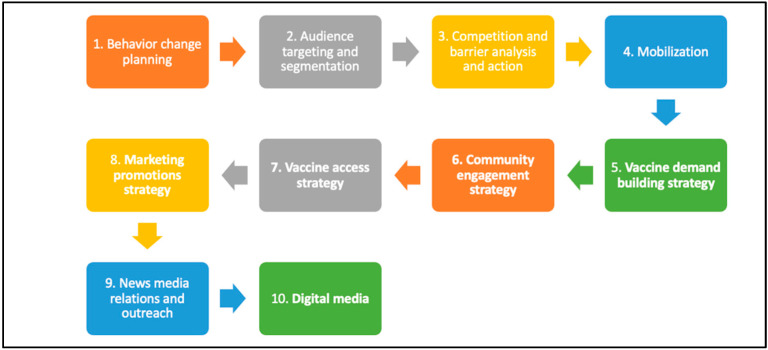
Guidelines for developing a pre-emptive COVID-19 vaccination uptake strategy (adapted from French et al., 2020 [24]).

**Figure 2 vaccines-11-00459-f002:**
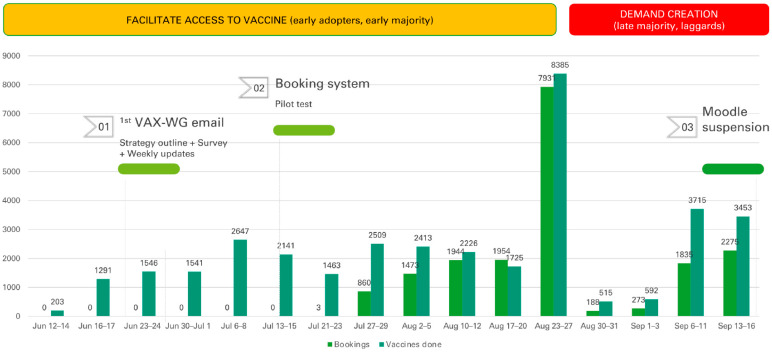
Number of vaccines administered and bookings, and campaign milestones.

## Data Availability

Not applicable.

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
