# Peer review of "Using Social Marketing to Promote COVID-19 Vaccination Uptake: A Case Study from the “AUBe Vaccinated” Campaign"

_vaccines, 2023, doi:10.3390/vaccines11020459_

Round 1

Reviewer 1 Report

This is a novel social topic regarding vaccination. In many countries due to lack of scientific updates, people reject vaccination mode of prevention. This problem is more sever where ancient tribal population exist. So, such studies are welcomed but in a deep manner. Having the data only in University people (people usually with high literate and educated mass) make the study to be limited. So, this kind of study is required in public sectors especially where education is not dominated the society.

So, a detailed literature search is needed to check detailed about the rejection of vaccination in general and covid-19 in particular. Statistical values to be incorporated. This is crucial to come up with a hypothesis for this study.

And the current study is need to be incorporated with public sector and comparison be made between both. 

Materials methods section is not very scientific. It seems an essay writing on a market survey and more appropriate to a journal of economics and social studies. I felt less science in the manuscript. What was the sampling method, inclusion, exclusion criteria etc. are largely missing.  Statistics is also missing.

Fig 1 and 2 are not really required.

Figure 3 is the only output of this study and has not enough scientific rigors.

In general the article is too long as per the data. It can be a communication and be written in maximum 3-4 printable pages excluding references.

So, I recommend the authors to make it a short communication and increase the sample size.

Author Response

This is a novel social topic regarding vaccination. In many countries due to lack of scientific updates, people reject vaccination mode of prevention. This problem is more sever where ancient tribal population exist. So, such studies are welcomed but in a deep manner. Having the data only in University people (people usually with high literate and educated mass) make the study to be limited. So, this kind of study is required in public sectors especially where education is not dominated the society. 

So, a detailed literature search is needed to check detailed about the rejection of vaccination in general and covid-19 in particular. Statistical values to be incorporated. This is crucial to come up with a hypothesis for this study.

And the current study is need to be incorporated with public sector and comparison be made between both.  

Response 1.1. Thank you for reviewing our manuscript and providing constructive feedback. We have addressed your requests as indicated below.

Based on Reviewer 2’s comments on the introduction and background, we believe we provided a sufficient and detailed literature review on the issue at hand and on the context where our study takes place. 

Materials methods section is not very scientific. It seems an essay writing on a market survey and more appropriate to a journal of economics and social studies. I felt less science in the manuscript. What was the sampling method, inclusion, exclusion criteria etc. are largely missing.  Statistics is also missing. 

Response 1.2. We followed the reporting style of case studies, a practice-oriented research practice generally used in business research. As the reviewer – but also some readers - might not be familiar with this methodology, we added a paragraph with references to the Case Study Methodology in Business Research. This seminal book describes this type of descriptive reporting, which is not intended to provide statistical inference or generalizations, but to describe a case to generate knowledge. The paragraph reads as follows:

“We adopt a practice-oriented research perspective described in the Case Study Methodology in Business Research [25]. Practice-oriented research aims to describe an intervention and reflect on its findings. These, in turn, contribute to enhancing practitioners’ knowledge and fostering the development of the field. As such, we do not intend to contribute to theory development or any other generalizations [25]. The structure of a practice-oriented case study follows the one of a descriptive, observational study. In this section, we discuss the context and methods and elaborate on the results in the discussion section.”

Fig 1 and 2 are not really required.

Response 1.3. Thank you for this suggestion. While we agreed that figure 2 (the campaign logo) was not essential, we retained figure 1 as it depicts the methodological framework we followed – which we believe is essential.

Figure 3 is the only output of this study and has not enough scientific rigors. 

Response 1.4. Figure 3 represents the distribution of vaccines over time, overlayed with the information about campaign events. This is not intended to represent precision in the measures, but to visualise the campaign’s impact. Therefore, we believe it is essential to be retained.

In general the article is too long as per the data. It can be a communication and be written in maximum 3-4 printable pages excluding references. 

So, I recommend the authors to make it a short communication and increase the sample size.

Response 1.5. The reviewer raises two points here. The first relates to the sample size. Unfortunately, the issue is not applicable because we did not undertake a trial nor tried to make generalisations based on the survey data we collected to inform the campaign strategy. The second point is about the manuscript length. We believe the manuscript is well articulated and provides a precise reflection on our experience. However, we are open to considering specific suggestions about where we could reduce the length of the article should the Editors believe it needs to be shortened.

Reviewer 2 Report

The purpose of this paper was to report the planning and implementation of a program in the Spring of 2021 at the American University of Beirut (AUB) in Lebanon to vaccinate the entire community by the beginning of the academic year, as the campus was open only to vaccinated individuals.  The Introduction of the paper is very informative and well done.

Authors, the Materials and Methods section is a bit confusing as you seem to combine methods, results and some discussion.  Prior to publication you will need to separate the results you mention in the Materials and Methods and put these in your Results section.  Also do not use this section for discussion and comparison of your results to other studies, this belongs in the Discussion.  If you review the paper with this separation in mind, you will determine how to separate the narrative.  For example what you describe in lines 189-198 seem to be results from your survey.  

Authors you did not present much data on the survey that you did.  It  would be appropriate to know how individuals responded to the survey questions, but there did not seem to much emphasis on the survey results.  Perhaps these results are within the Materials and Methods section.   For example what percentage of the population surveyed had to encouraged through the different methods to be vaccinated?

Authors the paper has some interesting observations on COVID vaccination acceptance or refusal in a very limited and intellectual population.  However, your presentation needs to have better separation of the various sections in the paper as mentioned above.

Author Response

The purpose of this paper was to report the planning and implementation of a program in the Spring of 2021 at the American University of Beirut (AUB) in Lebanon to vaccinate the entire community by the beginning of the academic year, as the campus was open only to vaccinated individuals.  The Introduction of the paper is very informative and well done.

Response 2.1. Thank you for your appreciative and constructive comments, which we welcomed very much. We have addressed your requests as indicated below.

Authors, the Materials and Methods section is a bit confusing as you seem to combine methods, results and some discussion.  Prior to publication you will need to separate the results you mention in the Materials and Methods and put these in your Results section.  Also do not use this section for discussion and comparison of your results to other studies, this belongs in the Discussion.  If you review the paper with this separation in mind, you will determine how to separate the narrative.  For example what you describe in lines 189-198 seem to be results from your survey.

Response 2.2. We thank the reviewer for his comment. We used a case study format, which I understand might need to be clarified. As also Reviewer 1 mentioned this issue, we included a small paragraph to outline the methods and refer to a seminal book so that the readers could appraise this methodology:

“We adopt a practice-oriented research perspective described in the Case Study Methodology in Business Research [25]. Practice-oriented research aims to describe an intervention and reflect on its findings. These, in turn, contribute to enhancing practitioners’ knowledge and fostering the development of the field. As such, we do not intend to contribute to theory development or any other generalizations [25]. The structure of a practice-oriented case study follows the one of a descriptive, observational study. In this section, we discuss the context and methods and elaborate on the results in the discussion section.”

Following your suggestions, many sections from the methods were moved to the results or discussion section. We are comparing are studies to the results of other studies in the discussion.    

Authors you did not present much data on the survey that you did.  It  would be appropriate to know how individuals responded to the survey questions, but there did not seem to much emphasis on the survey results.  Perhaps these results are within the Materials and Methods section.   For example what percentage of the population surveyed had to encouraged through the different methods to be vaccinated?

Response 2.3. We did not provide much data from the survey because it was not intended to generate research data but rather to inform the campaign strategy. In any case, we added the survey results to the results section, including the number of individuals who responded. As such, we reported the information that justified our segmentation approach and choice of communication channels. The survey was not intended to assess the intention to get vaccinated before the campaign to compare it after - as we did not undertake a formal research evaluation.

Authors the paper has some interesting observations on COVID vaccination acceptance or refusal in a very limited and intellectual population.  However, your presentation needs to have better separation of the various sections in the paper as mentioned above.

Response 2.4. Thank you for pointing this out. We have restructured the methods, results, and discussion section to reflect on different elements of the campaign (approach, population segmentation, strategy, and results). We hope that the manuscript now flows better and it is clearer.

Round 2

Reviewer 1 Report

A short manuscript always attract the readers, so try to reduce the as per the results.